# AUTO-REGRESSIVE NEXT-TOKEN PREDICTORS ARE UNIVERSAL LEARNERS

## ABSTRACT

Large language models display remarkable capabilities in logical and mathematical reasoning, allowing them to solve complex tasks. Interestingly, these abilities emerge in networks trained on the simple task of next-token prediction. In this work, we present a theoretical framework for studying auto-regressive next-token predictors. We demonstrate that even simple models such as linear next-token predictors, trained on Chain-of-Thought (CoT) data, can approximate any function efficiently computed by a Turing machine. We introduce a new complexity measure—length complexity—which measures the number of intermediate tokens in a CoT sequence required to approximate some target function, and analyze the interplay between length complexity and other notions of complexity. Finally, we show experimentally that simple next-token predictors, such as linear networks and shallow Multi-Layer Perceptrons (MLPs), display non-trivial performance on text generation and arithmetic tasks. Our results demonstrate that the power of today's LLMs can be attributed, to a great extent, to the auto-regressive next-token training scheme, and not necessarily to a particular choice of architecture.

## 1 INTRODUCTION

Large language models have achieved tremendous progress in various NLP tasks, such as machine translation, logical reasoning, coding and natural language understanding. These models, like GPT-3, GPT-4 and LaMDA (Brown et al., 2020; OpenAI, 2023; Thoppilan et al., 2022), are trained on massive amounts of text data and learn to generate coherent and contextually relevant responses to input prompts. Amazingly, such language models are mostly trained with a single objective: predicting the next token. While this objective seems extremely simplific, auto-regressive next-token predictors trained on rich enough data are able to solve strikingly complex tasks (Bubeck et al., 2023). This raises the question of whether such next-token predictors are merely "glorified" autocomplete models, which happened to memorize the entire internet, or are they truly performing novel logical reasoning. To this end, it has been shown that the ability of language models to compute complex functions can be greatly enhanced by using Chain-of-Thought (CoT) and scratchpad techniques (Wei et al., 2022b; Kojima et al., 2022; Lightman et al., 2023; Nye et al., 2021), allowing the models to perform unrestricted intermediate computations before arriving at a final answer.

In this work, we introduce a theoretical framework for studying auto-regressive next-token predictors. We demonstrate that much of the power of today's language models in logical reasoning can be attributed to the nature of the auto-regressive learning, and not to a particular choice of architecture. We show theoretically that very simple models trained to only predict the next token in an auto-regressive fashion can be used to solve extremely complex tasks when utilizing CoT techniques. In particular, we show that even linear predictors—models where the next-token probability is a linear function of the input sequence—are already powerful enough to compute *any Turing computable function*. The main theoretical result in the paper is captured in the following informal statement:

**Theorem 1** (informal). *For any function $f$ that can be* efficiently *computed using a Turing machine, there exists a dataset $D$ such that training a (linear) auto-regressive next-token predictor on $D$ results in a predictor that approximates $f$.*

That is, any computer program or intelligent agent that can be simulated by a computer, can be learned, given the right dataset, by a simple next-token predictor.

To understand the power of auto-regressive learning, observe that a result equivalent to Theorem 1 is not possible in the classical supervised learning setting, where the learner is given access only to the input sequence and the target label. It is well-known that no learning algorithm can efficiently learn the class of all (efficient) Turing computable functions (Valiant, 1984), given only the input and the output of the function (without access to intermediate supervision). In fact, in classical supervised learning, there are only a few function classes that are known to be *efficiently learnable*—function classes for which there exists a learning algorithm that can efficiently recover the target function given a labeled dataset. Learnable function classes are known to have fundamental limitations to their computational capacity. For example, the class of linear predictors is efficiently learnable in many settings, e.g. using the Perceptron algorithm (Rosenblatt, 1958). However, a famous result in Minsky & Papert (2017) shows that linear predictors cannot compute simple functions such as the XOR function. Auto-regressive learning, however, presents a striking difference. While linear next-token predictors are still *efficiently learnable* using simple algorithms such as SGD, their computational capacity greatly surpasses the capacity of their *classical* counterparts. Since auto-regressive inference introduces a sampling function[1] after each step, it allows linear next-token predictors to compute non-linear functions. As implied by Theorem 1, linear next-token predictors can implement practically any target function of interest.

While next-token predictors have the capacity to generate highly proficient learners, this does not come without a cost. One significant expense is the requirement to provide the learning model with potentially long sequences of tokens that detail the internal computations of the target. This requirement can be resource-intensive and often impractical. As such, it prompts the introduction of a new measure of learning complexity, analogous to sample complexity or run-time complexity: the *length complexity*. This type of complexity measures the quantity of intermediate tokens in a CoT necessary for the model to learn a particular concept class. We explore this complexity in the context of the parity learning problem, an extension of the XOR problem that is known to be computationally hard to learn in some settings. We demonstrate how traditional forms of complexity, such as sample or run-time complexity, can be traded off with length complexity when learning parities. Specifically, we show that an *increase* in the complexity of the hypothesis class—and therefore in sample or computational complexity—leads to a *decrease* in length complexity. This opens up a new path for the theoretical investigation of auto-regressive learning, by studying the interplay between these different complexity measures.

To substantiate our theoretical results, we experimentally illustrate the power of auto-regressive learning in enhancing the performance of simple models. We train a linear next-token prediction network on the TinyStories dataset (Eldan & Li, 2023), a collection of short stories composed of simple words. We observe that linear models, once trained on this dataset, frequently generate plausible and grammatically sound stories. Next, we demonstrate that a shallow Multi-Layer Perceptron (MLP) with 775M parameters (no attention layers), can learn to correctly multiply two 4-digit numbers, given CoT data. Our MLP outperforms GPT-4 in this task, and achieves comparable results to Goat, a 7B-parameter transformer trained to solve arithmetic tasks (Liu & Low, 2023).

## 1.1 RELATED WORK

**CoT Reasoning** The proposition of supervising intermediate logical steps as an effective approach for problem-solving is well established, predating the advent of Transformer models. The technique was found to be particularly beneficial in solving arithmetic problems (Roy & Roth, 2016). This idea became very popular with the introduction of the Chain-of-Thought (CoT) approach, where models are prompted to elucidate their thought process prior to yielding a final outcome (Wei et al., 2022b; Kojima et al., 2022; Lightman et al., 2023). Recent developments have further demonstrated the efficacy of the CoT method in the training of smaller student models (Li et al., 2023; 2022; Magister et al., 2022). Another method that bears similarity to CoT is the "scratchpad" technique, which allows models to record intermediate computations that subsequently aid in deriving the final answer (Nye et al., 2021). Such techniques have been shown to enhance performance across a variety of logical reasoning and arithmetic tasks. The research presented in this paper aims to contribute to the theoretical understanding of CoT reasoning in auto-regressive models. Our work illustrates how the employment of CoT can significantly amplify the capabilities of simple models. Furthermore, we introduce a novel complexity measure, the *length complexity*, that allows us to study the influence

---

[1] In our analysis we focus on the zero-temperature/argmax sampling, which acts as an explicit non-linearity.

of the length of the intermediate sequence of tokens within CoT on the difficulty of the learning problem.

**Language Models for Arithmetic Tasks**   Leveraging large language models to tackle mathematical reasoning and arithmetic tasks has gained significant interest, a trend that is discussed at length in a recent survey (Lu et al., 2022). While these models have demonstrated a promising capacity for solving an array of mathematical problems, they often encounter difficulties in executing straightforward arithmetic operations, such as the multiplication and addition of large numbers (Nogueira et al., 2021; Qian et al., 2022). Previous studies have suggested that the efficiency of language models in arithmetic tasks can be dramatically enhanced by structuring them to perform calculations using an algorithmic pipeline, facilitating step-by-step execution (Muffo et al., 2023). A notable contribution in this realm is the recent work by Liu & Low (2023), where they fine-tuned a moderately sized (7B-parameter) transformer employing the CoT method to perform complex arithmetic operations, including the multiplication of large numbers—a challenge even for advanced models like GPT-4. A very recent work studies the ability of small transformers trained from scratch to solve arithmetic tasks (Lee et al., 2023). In our study, we further substantiate this claim by demonstrating that a small MLP, devoid of any attention mechanism, can match the performance of the transformer in Liu & Low (2023) in 4-digit multiplication, provided that it receives appropriate intermediate supervision. This highlights that the capability of language models for arithmetic and mathematical reasoning is largely attributable to the CoT and next-token prediction techniques, rather than the specific architectural choice.

**Beyond Transformers**   Although the transformer architecture (Vaswani et al., 2017) currently stands as the leading approach in language modeling, it is noteworthy that a diverse range of other architectures have served this purpose over time. A notable instance is the application of Recursive Neural Networks (RNNs) (Hochreiter & Schmidhuber, 1997), a model highly popular for language modeling only a few years back, due to its efficient and inherent sequence processing capabilities (Mikolov et al., 2010). Furthermore, convolutions have also been explored for language modeling tasks (Dauphin et al., 2017). A work more related to our own leveraged linear dynamical systems to model text (Belanger & Kakade, 2015). Recent years have witnessed an emerging interest in substituting the attention layer of transformers, primarily due to its high computational cost, with simpler and more efficient alternatives. In this vein, the work of Katharopoulos et al. (2020) introduced the linear transformer, where the attention layer was replaced with a more computationally-friendly linear layer. Concurrently, Zhai et al. (2021) advanced an Attention-Free Transformer. More recent advancements include the RWKV architecture (Peng et al., 2023), a modern variant of the RNN architecture inspired by transformers, which exhibits competitive performance when trained on large datasets. Some studies have proposed the use of simpler MLP-based architectures as feasible alternatives to transformers (Tolstikhin et al., 2021; Liu et al., 2021). Our work contributes to this ongoing discourse by conducting both theoretical and empirical investigations into the potential of very simple models, such as linear models and small MLPs, training them to solve complex tasks by leveraging the power of next-token auto-regressive learning.

**Related Theoretical Work**   Despite the rapid pace of practical advancements in the realm of language models and transformers, the theoretical underpinning remains comparatively unexplored. Early investigations have established the universality of transformers (i.e., their ability to emulate any Turing machine) given the incorporation of a recurrent module (Yun et al., 2019; Wei et al., 2022a). More recently, it has been demonstrated that transformers can simulate universal computers when incorporated into an execution loop (Giannou et al., 2023). The work of Liu et al. (2022) shows that Transformers can simulate Automata, which are equivalent to bounded-memory programs, using surprisingly few layers. Turing universality extends to other language modeling architectures, such as RNNs Siegelmann & Sontag (1992). A study by Edelman et al. (2022) underscores the inductive biases of self-attention, demonstrating that bounded-norm Transformer networks can represent sparse functions with logarithmically scaling sample complexity. The work of Feng et al. (2023) theoretically demonstrates the importance of CoT for solving mathematical problems with transformers. Of particular relevance to our study is the work of Wies et al. (2022), which delves into how sub-task decomposition and the CoT technique can facilitate the learning of computationally challenging problems. Similarly to our study, Wies et al. (2022) also explores parity learning with intermediate supervision and demonstrates that arbitrary Turing machines can be efficiently learned

by language models trained with CoT. Our work extends these findings, introducing a theoretical framework that enables broader examination of auto-regressive learning. We show that even linear predictors can efficiently learn Turing computable functions. In addition, our results offer improved length complexity bounds for learning parities, indicating that parities can be learned using $O(\log n)$ intermediate tokens, a marked reduction from the $O(n)$ intermediate tokens in Wies et al. (2022).

## 2 THEORY

The key principle in our theoretical results is the differentiation between "classical" supervised learning and Auto-Regressive (AR) learning. In supervised learning, there is a clear separation between the input and the label (or target). The learner gets a dataset of inputs with their labels, and needs to find a model that correctly predicts the label of a new input example. While supervised learning tasks can sometimes be easy (e.g., when the label is given by a linear function of the input features), this task becomes very hard, or even impossible, when the function used for generating the labels requires a complex computational process (Valiant, 1984). This hardness stems from the fact that the internal computation is not available to the learner, who only observes the input and the corresponding final output.

In AR learning, on the other hand, the situation is different. AR learners get a sequence of tokens, and treat every token both as an input (for predicting future tokens) and as a label (for sequences of previous tokens). Coupling AR learning with the chain-of-thought technique results in a learning paradigm where the internal computations required for reaching the final answer become available to the learner both as inputs and as *labels*. This naturally allows supervision on intermediate steps in the computation/reasoning process, which greatly simplifies the learning task.

In the following sections we detail our theoretical results. In Section 2.1 we formally define the framework of AR Learning and Learnability, in an analogous way to classical PAC Learning. We then show how PAC Learnable hypothesis classes can be used for constructing AR Learnable classes, and discuss the special case of linear classes (which are known to be efficiently PAC Learnable). In Section 2.2 we discuss approximation results, namely understanding what types of function a given AR model can compute. To this end, we consider the function computed by the model to be the function mapping the input tokens to the final token(s), allowing the model to arbitrarily use internal computations in a chain-of-thought manner. Following this, we show that even linear AR models can compute very complex functions, for example emulating arbitrary Turing machines. Finally, in Section 2.3 we introduce *length complexity*, which measures how many intermediate tokens are required in order to learn to compute a given function. We show that using more intermediate tokens, i.e. increasing the length complexity, can reduce time/sample complexity, and vice-versa.

### 2.1 LEARNABILITY RESULTS

Let $\mathbb{D}$ be a finite set of tokens, let $\mathcal{X} = \mathbb{D}^n$ be the space of contexts of $n$ tokens, and let $\mathcal{Z} = \mathbb{D}^*$ be a space of strings of tokens. For some $t$, we denote $\mathcal{Z}_t = \mathbb{D}^t$. An Auto-Regressive (AR) function $h$ is a mapping $\mathcal{X} \times \mathcal{Z} \to \mathbb{D}$ (we assume a deterministic function). An AR hypothesis class $\mathcal{H}$ is a set of AR functions. Fix some $T \in \mathbb{N}^2$. For some distribution $\mathcal{D}$ over $\mathcal{X} \times \mathcal{Z}_T$, we say that $\mathcal{D}$ is *realizable* by the AR class $\mathcal{H}$ if there exists a function $h \in \mathcal{H}$ such, with probability 1 over $(\boldsymbol{x}, \boldsymbol{z}) \sim \mathcal{D}$, we have $h(\boldsymbol{x}, \boldsymbol{z}_{<t}) = z_t$ for all $t \leq T$ (where $\boldsymbol{z}_{<t}$ denotes the first $t-1$ coordinates of $\boldsymbol{z}$). In other words, the pair $(\boldsymbol{x}, \boldsymbol{z})$ is realizable by $h$ if $h$ accurately predicts the next token for all prefixes $\boldsymbol{z}_{<t}$ of $\boldsymbol{z}$. We now define Learnability in the AR framework:

**Definition 2.** *We say that $\mathcal{H}$ is* AR Learnable *if there exists a function* $m : (0,1)^2 \to \mathbb{N}$ *and an algorithm such that for every* $\epsilon, \delta \in (0,1)$ *and distribution $\mathcal{D}$ realizable by $\mathcal{H}$, given a sample of size $m(\epsilon, \delta)$ from $\mathcal{D}$, returns with probability (w.p.)* $\geq 1 - \delta$ *a function $\hat{h} \in \mathcal{H}$ s.t.* $\Pr\left[\exists t \leq T \text{ s.t. } \hat{h}(\boldsymbol{x}, \boldsymbol{z}_{<t}) \neq z_t\right] \leq \epsilon$. *Furthermore, we say that $\mathcal{H}$ is* efficiently AR Learnable *if it is* AR Learnable *with an algorithm running in polynomial time.*

That is, a class $\mathcal{H}$ is (efficiently) AR Learnable if there exists an (efficient) algorithm that finds, w.h.p., a next-token predictor with low error.

---

[2]In Section 2.3 we study how the choice of $T$ affects the complexity of the learning problem, but for now we treat $T$ as a fixed parameter of the learning problem.

We now show that hypothesis classes that are learnable in the classical sense (i.e., by supervised learning), naturally induce hypothesis classes that are AR Learnable. Let $\mathcal{H}$ be some AR hypothesis class. We assume that $\mathcal{H}$ can be decomposed into "standard" hypothesis classes in the following sense. Let $\{\mathcal{H}_t\}_{t=1}^{\infty}$ be a sequence of classes, where $\mathcal{H}_t$ is a class of functions $\mathcal{X} \times \mathcal{Z}_{t-1} \mapsto \mathbb{D}$. We assume that $\mathcal{H} = \mathcal{H}_1 \times \mathcal{H}_2 \times \ldots$. Namely, we associate every $h \in \mathcal{H}$ with a sequence $(h_1, h_2, \ldots)$, where $h_i \in \mathcal{H}_i$, s.t. for every $\boldsymbol{x} \in \mathcal{X}$ and $\boldsymbol{z} \in \mathcal{Z}_{t-1}$ we have $h(\boldsymbol{x}, \boldsymbol{z}_{<t}) = h_t(\boldsymbol{x}, \boldsymbol{z}_{<t})$. While we define $\mathcal{H}$ on arbitrarily long sequences, when we study learnability we limit ourselves to discussing sequences of length at most $T$. In particular, we can assume $\mathcal{H} = \mathcal{H}_1 \times \cdots \times \mathcal{H}_T$. The following result shows that PAC Learnability of the underlying hypothesis classes (as defined e.g. in Shalev-Shwartz & Ben-David (2014)) implies AR Learnability of the class $\mathcal{H}$:

**Theorem 3.** *If $\mathcal{H}_1, \ldots, \mathcal{H}_T$ are (efficiently) PAC Learnable with sample complexity $m(\epsilon, \delta)$, then $\mathcal{H} = \mathcal{H}_1 \times \cdots \times \mathcal{H}_T$ is (efficiently) AR Learnable with sample complexity $m(\epsilon/T, \delta/T)$.*

The proof (in Appendix A) is a simple reduction using the standard notion of PAC Learnability.

**Linear Decoder**

From Theorem 3, efficiently learnable classes induce classes that are efficiently learnable in the Auto-Regressive setting. For example, by letting $\mathcal{H}_t$ be a class of linear functions, we can use known results on learning linear classifiers to show that the induced AR hypothesis class is efficiently learnable. We define the linear AR hypothesis class as follows.

**Definition 4.** *Let $\psi : \mathbb{D} \to \mathbb{R}^d$ be some embedding of the dictionary. With some abuse of notations, for $\boldsymbol{z} \in \mathbb{D}^t$ we define $\psi(\boldsymbol{z}) = [\psi(z_1), \ldots, \psi(z_t)] \in \mathbb{R}^{d \times t}$. Fix some $t$, let $\boldsymbol{W} \in \mathbb{R}^{\mathbb{D} \times d \times (n+t)}$, and for all $\boldsymbol{x} \in \mathcal{X}$ and $\boldsymbol{z} \in \mathcal{Z}_t$ define $h_{\boldsymbol{W}}(\boldsymbol{x}, \boldsymbol{z}) = \arg\max_{D \in \mathbb{D}} \langle W_D, \psi([\boldsymbol{x}, \boldsymbol{z}]) \rangle$. Denote the function class of all linear predictors $\mathcal{H}_t^{\mathrm{Lin}} = \{h_{\boldsymbol{W}} : \boldsymbol{W} \in \mathbb{R}^{\mathbb{D} \times d \times (n+t)}\}$.*

Observe that the class $\mathcal{H}_t^{\mathrm{Lin}}$ is PAC-learnable in polynomial time. Under some margin conditions and using a convex surrogate loss function, this class is in fact learnable using SGD (Shalev-Shwartz & Ben-David, 2014). Therefore, for the linear AR hypothesis class $\mathcal{H}^{\mathrm{Lin}} = \mathcal{H}_1^{\mathrm{Lin}} \times \cdots \times \mathcal{H}_T^{\mathrm{Lin}}$, we get that $\mathcal{H}^{\mathrm{Lin}}$ is efficiently learnable in the Auto-Regressive setting.

## 2.2 Approximation Results

We showed that when the AR hypothesis class $\mathcal{H}$ is induced from a sequence of (efficiently) learnable hypothesis classes, then $\mathcal{H}$ is also (efficiently) AR learnable. In particular, $\mathcal{H}^{\mathrm{Lin}}$ is efficiently AR learnable, as a product of linear classes. We now show that while learnability transfers from the classical setting to the AR setting, in AR learning we can get much stronger *approximation* guarantees. In fact, while linear classes are relatively limited in the standard setting, we show that the linear AR class $\mathcal{H}^{\mathrm{Lin}}$ is extremely powerful. Namely, we show that linear AR functions can efficiently approximate any Turing computable function.

We first need a proper definition of what are the functions that AR hypotheses "compute". For some AR hypothesis $h$, define the output of the auto-regression process at time $t$ to be $h^{(t)}(\boldsymbol{x})$, defined recursively by: $h^{(1)}(\boldsymbol{x}) = h(\boldsymbol{x}, \emptyset)$, and $h^{(t)}(\boldsymbol{x}) = h\left(\boldsymbol{x}, \left(h^{(1)}(\boldsymbol{x}), \ldots, h^{(t-1)}(\boldsymbol{x})\right)\right)$. For now, we focus on AR hypotheses that are evaluated for $T$ steps, for some fixed $T \in \mathbb{N}$. In Section 2.3 we discuss how the choice of $T$ (length complexity) interacts with different measures of complexity. We define the function computed (approximated) by $h$ as follows:

**Definition 5.** *Fix some target $f : \mathbb{D}^n \to \mathbb{D}$ and some AR hypothesis $h$. Then, we say that $h$ computes $f$, if for every input $\boldsymbol{x} \in \mathbb{D}^n$ we have $h^{(T)}(\boldsymbol{x}) = f(\boldsymbol{x})$. Additionally, for some distribution $\mathcal{D}$ over $\mathbb{D}^n$, we say that $h$ $\epsilon$-approximates $f$ w.r.t. $\mathcal{D}$, if $\Pr_{\mathcal{D}}\left[h^{(T)}(\boldsymbol{x}) \neq f(\boldsymbol{x})\right] \leq \epsilon$.*

In other words, we say that $h$ computes $f$ if after running auto-regression for $T$ steps, it outputs a value that agrees with $f$. Note that we ignore all the intermediate outputs of $h$ and observe only the final output. This is in alignment with common practice, where we let language models use arbitrarily long chain-of-thought/scratchpad before arriving at the final answer[3].

---

[3]Here we assume that $f$ outputs a single token in $\mathbb{D}$, and therefore observe only the last token produced by the auto-regression. However, we note that this can be extended to the case where $f$ outputs multiple tokens, and we observe a sequence of tokens at the end of the auto-regression.

Next, we show that if some AR class $\mathcal{H}$ is learnable, then auto-regressive learning of distributions realizable by $h \in \mathcal{H}$ returns an approximator for the function computed by $h$:

**Theorem 6.** *Assume that $\mathcal{H}$ is (efficiently) AR Learnable with sample complexity $m(\epsilon, \delta)$. Then, there is an (efficient) algorithm s.t. for any $\epsilon, \delta$ and distribution $\mathcal{D}$ realizable by some $h \in \mathcal{H}$, given a sample of size $m(\epsilon, \delta)$, returns w.p. $\geq 1 - \delta$ a function $\hat{h}$ s.t. $\hat{h}^{(T)}$ $\epsilon$-approximate $h^{(T)}$ w.r.t. $\mathcal{D}$.*

The proof follows by induction from the definitions (see Appendix A). Theorem 6 shows that using AR learning, we can learn to approximate the function computed by the underlying AR function $h$.

**Approximation Capacity of Linear Hypotheses**

We now limit ourselves to a dictionary with only two tokens $\mathbb{D} = \{0, 1\}$, to be compatible with standard analysis of computations with Boolean inputs/outputs. We will show that linear AR functions can approximate a very large class of functions—namely, the class of *linear threshold circuits*.

**Definition 7.** *A linear threshold function is a func. of the form $x \mapsto \sigma(\langle \boldsymbol{w}, x \rangle + b)$ for $\sigma(x) = \mathbf{1}_{x \geq 0}$. A linear threshold circuit is a Boolean circuit where every gate computes a linear threshold function.*

The following result shows that linear AR functions can approximate any linear threshold circuit:

**Theorem 8.** *Assume that $f : \{0, 1\}^n \to \{0, 1\}$ can be computed by a linear threshold circuit with at most $T$ gates. Then, $f$ can be computed by a linear AR function $h \in \mathcal{H}^{\mathrm{Lin}}$.*

The proof of the above result uses the fact that a linear threshold function can be implemented using $\arg\max$ over a linear function, in the case where $\mathbb{D} = \{0, 1\}$ (full proof in Appendix A).

We note that any Turing computable function can be computed by a linear threshold circuit of some size $T$ that scales polynomially with the runtime of the Turing machine (see e.g. Arora & Barak (2009)). Therefore, we get that linear AR functions can compute any Turing computable function, with only polynomial blow-up in run-time. This leads to the following result:

**Corollary 9.** *For any function $f$ that is Turing computable in time $T(n)$, and for any distribution $\mathcal{D}$ over inputs of size $n$, there exists a dataset of strings of tokens, each of size $\mathrm{poly}(T(n))$, s.t. training a linear AR model over this dataset efficiently recovers a function that approximates $f$ w.r.t. $\mathcal{D}$.*

To prove the above, we consider a dataset generated by a linear model simulating the target Turing machine which computes $f$.

## 2.3 Length Complexity

We showed that even simple classes like linear AR predictors can approximate any Turing computable function. Since linear predictors can be learned efficiently, we get a learning scheme that can efficiently learn virtually any function of interest. This is in contrast with the standard supervised learning setting, where efficiently learnable function classes are typically very limited in their expressive power. However, we note that the complexity of learning did not magically "disappear". To make learning possible, we require that the learner has, during learning, access to a sequence of tokens representing the internal CoT generated by the target it aims to imitate. While the length of this sequence is still reasonable (polynomial in the problem parameters), acquiring data with such long sequences might be costly, or even impossible.

In this section we introduce *length complexity*, a new notion of learning complexity that quantifies the number of intermediate tokens required for learning some concept class, i.e. the length of the CoT supervision provided to the model during training. The length complexity complements common complexity measures such as sample and run-time complexity, and we show that in some cases we can trade off sample/computational complexity for length complexity, and vice versa.

We begin with a formal definition of *length complexity*. Fix some distribution over $\mathbb{D}^n$, some AR hypothesis class $\mathcal{H}$ and some target concept class $\mathcal{F}$ of functions $\mathbb{D}^n \to \mathbb{D}$. The definition below extends Definition 5 to function classes, which allows an explicit discussion on length complexity.

**Definition 10.** *We say that $\mathcal{H}$ computes $\mathcal{F}$ with length complexity $T$, if $T$ is the minimal number satisfying that for every $f \in \mathcal{F}$ there exists some $h \in \mathcal{H}$ such that, for all $\boldsymbol{x} \in \mathbb{D}^n$ we have $h^{(T)}(\boldsymbol{x}) = f(\boldsymbol{x})$. Additionally, we say that $\mathcal{H}$ $\epsilon$-approximates $\mathcal{F}$ with length complexity $T$ if for every $f \in \mathcal{F}$ there exists some $h \in \mathcal{H}$ s.t. $\mathrm{Pr}_{\mathcal{D}}\left[h^{(T)}(\boldsymbol{x}) \neq f(\boldsymbol{x})\right] \leq \epsilon$.*

From Theorem 8 we get that the class of linear threshold circuits of size $T$ can be $\epsilon$-approximated using linear AR functions with *length complexity* $T$. For small circuits this might not be an issue, but otherwise this dependence may be problematic. We expect that taking a richer AR hypothesis class $\mathcal{H}$ would result in reduction of the length complexity. In the rest of this section, we discuss the interplay between the choice of the AR hypothesis class and the different measures of complexity that it induces: sample complexity, computational complexity and length complexity.

**Length Complexity of Parities**

To demonstrate a concrete analysis of length complexity, we consider the well-studied problem of learning parities, a natural extension of the XOR problem (Minsky & Papert, 2017). In the parity learning problem, the inputs are sequences of $n$ bits, and the label is determined by the parity of the sum of an unknown subset of bits from the input. This problem is known to be computationally hard in some settings. For example, Statistical Query (SQ) algorithms and variants of gradient-descent need $\Omega(2^n)$ steps to solve the problem (Kearns, 1998; Shalev-Shwartz et al., 2017; Abbe & Sandon, 2018; Malach & Shalev-Shwartz, 2022), and it is hard to solve with limited memory (Raz, 2018).

We now formally define the set of parity functions. Assume $\mathbb{D} = \{0, 1\}$ (Boolean inputs). For some subset $A \subseteq [n]$, define the parity function over $A$ by $\chi_A(\boldsymbol{x}) = \sum_{i \in A} x_i \mod 2$. Let $\mathcal{P}_n$ be the class of all parity functions, $\mathcal{P}_n = \{\chi_A : A \subseteq [n]\}$. It is known that parities can be computed using $O(\log n)$ size linear threshold circuit (Kautz, 1961). So, Theorem 8 implies that a linear AR model can compute any parity function with logarithmic length complexity:

**Theorem 11.** *The class $\mathcal{P}_n$ can be computed using $\mathcal{H}^{\mathrm{Lin}}$, with length complexity $O(\log n)$.*

Since we showed that linear AR functions are efficiently learnable (Theorem 3), the above theorem implies that parities become efficiently learnable given $O(\log n)$ intermediate tokens. This is in contrast to the standard supervised learning setting, where linear functions cannot approximate parities (Daniely & Malach, 2020). We note that a similar result on learning parities with intermediate tokens appears in Wies et al. (2022), but with $O(n)$ length complexity (instead of $O(\log n)$).

We next show that by taking more complex hypothesis classes we can reduce the length complexity of computing $\mathcal{P}_n$. However, this comes at a cost of increasing either the sample or the computational complexity. We define a sequence of AR classes of growing complexity for computing $\mathcal{P}_n$. For every $k \leq n$, let $\mathcal{P}_{n,k}$ be the class of parities over subsets of size $\leq k$, namely $\mathcal{P}_{n,k} = \left\{ \chi_A : A \in \binom{[n]}{\leq k} \right\}$. The larger $n$ and $k$ are, the harder it is to learn $\mathcal{P}_{n,k}$ (via supervised learning). In particular, there are known lower bounds on learning $\mathcal{P}_{n,k}$ using Statistical Query (SQ) algorithms, a large family of algorithms that include variants of gradient-based learning algorithms (Blum et al., 2003). Roughly speaking, learning $\mathcal{P}_{n,k}$ using SQ algorithms requires run-time of $\binom{n}{\leq k} = O((n/k)^k)$, and the sample complexity of $O(k \log n)$. We define $\mathcal{H}^{(k)} = \mathcal{P}_{n,k} \times \mathcal{P}_{n+1,k} \times \ldots$, and show the following:

**Theorem 12.** *$\mathcal{H}^{(k)}$ can compute $\mathcal{P}_n$ with length complexity $\Theta(n/k)$.*

To prove the above result, we show that any parity over $n$ bits can be computed by constructing a "tree" of $k$-order parities, which reduces the length complexity by a factor of $k$ (see Appendix A). This decrease in length complexity comes at the cost of increasing the computational complexity of the learning *exponentially* with $k$ (for SQ algorithms and variants of GD). While the exact interplay between computational and length complexity depends on the learning problem, this result shows that sometimes decreasing the length complexity makes the problem computationally hard to learn. We believe that a fundamental understanding of the length complexity of different problems will allow us to better understand AR predictors. For example, discovering an intrinsic complexity measure for hypothesis classes (analogous to VC dimension or SQ dimension) that can be used to derive length complexity bounds is of particular interest. We leave such an investigation to future research.

## 3 EXPERIMENTS

We now turn to empirically validate our theoretical results, showing that very simple models perform surprisingly well when trained auto-regressively to perform next-token prediction. We start by training a simple linear model on a dataset of short stories, and then evaluate the performance of a small MLP on a task of arithmetic computations.

## 3.1 TINY STORIES

We test the efficiency of linear AR models on the TinyStories dataset (Eldan & Li, 2023), a synthetic dataset of short stories containing simple words. We train a linear model with context length of $T = 64$ on this dataset. The model has only three layers: 1) a standard (linear) embedding layer, mapping tokens into a vector of dimension $d = 256$; 2) a linear layer mapping $d \times T$ to $d \times T$ (using standard masking for next-token prediction during training); 3) an output embedding layer mapping vectors of dimension $d = 256$ back into the output space of all tokens (see Figure 1). To allow next-token

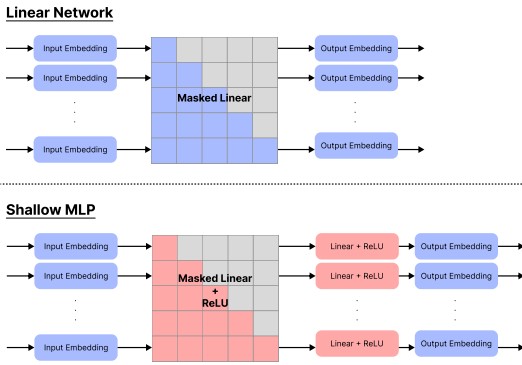

Figure 1: Illustration of the linear network and the MLP used in our experiments.

prediction training, we apply masking on the second linear layer, so that each output token only has access to previous tokens in the sequence. While the resulting classifier is linear, we note that this model is not exactly the linear AR model analyzed previously, as we allow sharing some parameters (namely, the input/output embedding parameters) across the different sequence positions. However, this is a close proxy to the idealized linear model. The model is optimized with the cross-entropy loss, using a softmax operation applied to the outputs. Altogether, the resulting model has roughly 162M active parameters. The model is trained for 5½ hours on a single A100 machine.

While the results are certainly inferior in quality to transformer-based language models[4], we note that the linear predictor often does produce coherent text. Below we show some example for prompts and the resulting output of the model. Notice that there are some grammatical errors (e.g. Prompt #2) or coneptual errors (e.g. Prompt #3), but the overall behavior seems reasonable.

| | |
|---|---|
| **Prompt #1:** | He was very scared |
| **Output:** | *He was very scared. He cried out loud, but then he heard a voice. He was afraid of the dark monster. He ran to the bear. He had a pain in his hand. He screamed and ran away. He was never seen again. He was scared of the monster and he went on. He never...* |
| **Prompt #2:** | Adam was hungry, so |
| **Output:** | *Adam was hungry, so he wanted to eat the dessert. But he knew it was  not a good at his sister . He thought, "Maybe I can eat this food, but I don't want to share with you."* |
| **Prompt #3:** | Alice was tired, so |
| **Output #1:** | *Alice was tired,  so she decided to take a nap . She put a blanket on the ground and started to cry. Then, she heard a noise. It was a nearby tree.* |
| **Output #2:** | *Alice was tired,  so she decided to go on an adventure . She hopped on the way to go home and look for her...* |

## 3.2 MULTIPLICATION

We now turn to demonstrate the power of next-token prediction with CoT reasoning for arithmetic tasks. We focus on the task of multiplying two 4-digit numbers, which has been shown to be challenging even for huge language models such as GPT-4 (Liu & Low, 2023). For this task, we train a simple Multi-Layered Perceptron (MLP) with four layers: 1) a standard (linear) embedding layer, from tokens to dimension $d = 128$; 2) a linear layer with a ReLU activation, applied across all the context window, mapping the input of $d \times T$ to an output of $d \times T$ (where we use a context length of $T = 307$); 3) a linear layer with a ReLU activation applied per token, mapping from $d$ to $d$; 4) a final output embedding, mapping back to the space of all tokens (see Figure 1). Similarly to the linear network, we mask future positions in the second layer. We note that while this network has non-linearity (unlike the previous model), it is still very simple compared to standard transformer-based networks (e.g., we use no attention mechanism). Altogether, our MLP has 775M active parameters.

Recently, a paper by Liu & Low (2023) instroduced Goat, a relatively small transformer fine-tuned from the LLaMA model that was able to outperform GPT-4 in various arithmetic tasks, when trained

---

[4]For comparison, in our experiments GPT-2 Small (124M parameters) reaches perplexity of 2.2 on the TinyStories datasets, where our linear model reaches a perplexity of 3.4 when trained in the same scheme.

| Prompt: | $1394 \times 8618 =$ |
|---|---|
| **MLP:** | $(4 \times 1 + 9 \times 10 + 3 \times 100 + 1 \times 1000) \times$ $(8 \times 1 + 1 \times 10 + 6 \times 100 + 8 \times 1000) =$ $\vdots$ `12013492` |
| **GPT-4:** | The multiplication of 1394 and 8618 equals `12` , `01` `4` , `05` `2` . |
| **Answer:** | **12013492** |

| Model | Acc. (exact/per-digit) |
|---|---|
| **MLP-775M** | 96.9% / 99.5% |
| **GPT-3.5** | 1.2% / 61.9% |
| **GPT-4**\* | 5.3% / 61.8% |
| **Goat-7B**\* | 96.9% / 99.2% |

Figure 2: **Left:** Comparison between the output of our MLP and GPT-4 on the 4-digit multiplication task (see full output in Appendix B). **Right:** Performance of GPT vs. MLP model on the 4-digit multiplication task. \*For GPT-4 and Goat-7B, we use the numbers as repored in Liu & Low (2023).

on data with intermediate calculations. We follow a similar procedure for training our model on 4-digit multiplication, with some key differences. First, we give more intermediate steps than in Liu & Low (2023), essentially unfolding the multiplication algorithm in the training sequences (see Figure 3.2). Second, we use a custom tokenization scheme, where we tokenize separately single digits $(1, 2, 3, \dots)$, signs $(\times, +, =)$ and also pairs of digits with multiplication sign $(1 \times 2, 3 \times 5,$ etc). This tokenization allows the model to quickly solve the single-digit multiplication task (by mapping pairs of multiplied digits to their product), which is a crucial tool in the multiplication algorithm. Finally, we also add zero-padding to some of the numbers, to get all strings to have the same length.

We split all pairs of 4-digit numbers arbitrarily, use $75\%$ for training, and keep the rest for validation. The network is trained from scratch for 17 hours on a single A100 GPU, going over 100M sequences (307M tokens) sampled uniformly from the training set. In Table 3.2 we compare the performance of our simple MLP (evaluated on 1000 validation examples) with GPT-3.5 (evaluated on the same examples), as well as to GPT-4 and Goat-7B on the same task (as reported in Liu & Low (2023)). We report both accuracy of the exact match of the final answer, as well as accuracy of individual digits in the final number. We note that the performance of our MLP matches the performance of the much larger fine-tuned transformer in Liu & Low (2023), and outperforms both GPT-3.5 and GPT-4 on this task. This demonstrates again that a lot of the power of language models can be attributed to the next-token auto-regressive training, and not necessarily to a particular architectural choice.

## 4 DISCUSSION

The emerging capabilities of large language models has triggered an ongoing debate about their potential and implications. Certain proponents assert that we are close to achieving Artificial General Intelligence (AGI), pointing to models such as GPT-4 which have already demonstrated perceived "sparks of AGI" (Bubeck et al., 2023). They argue that AGI is just a matter of scaling up—creating larger models, feeding them with more data, and increasing training time. In stark contrast, others dismiss these large models as merely sophisticated autocomplete systems, voicing concerns about their propensity to potentially absorb and perpetuate biased and harmful data Bender et al. (2021).

While this debate is far from settled, we hope that our work sheds light on the theoretical possibilities inherent in training auto-regressive next-token predictors. Our findings indicate that, given suitable data, simple next-token predictors can be trained to effectively learn virtually any function of interest. Consequently, if there exists some computer program capable of realizing AGI, then it is theoretically plausible to attain AGI through training simple next-token predictors, given the appropriate data. Admittedly, these assertions, in their current form, are somewhat theoretical, with practical application requiring data composed of potentially very long sequences of intermediate computations. However, we show that by modifying the choice of the hypothesis class we can possibly shorten the required sequence length, making our results more realistic. Therefore, we believe that our research can contribute towards a better, more nuanced understanding of both the capabilities and constraints associated with next-token predictors.

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

# A PROOFS

*Proof of Theorem 3.* Let $\mathcal{D}$ be some distribution over $\mathcal{X} \times \mathcal{Z}_T$ realizable by $\mathcal{H}$, and let $\mathcal{D}_t$ be the distribution over $(\mathcal{X} \times \mathcal{Z}_{t-1}) \times \mathbb{D}$, where we sample $(\boldsymbol{x}, \boldsymbol{z}) \sim \mathcal{D}$ and observe $((\boldsymbol{x}, \boldsymbol{z}_{<t}), z_t)$. Therefore, $\mathcal{D}_t$ is a labeled distribution realizable by $\mathcal{H}_t$, and so we can use a learner for $\mathcal{H}_t$ to find using $m(\epsilon/T, \delta/T)$ samples, with probability $1 - \delta/T$, a hypothesis $\hat{h}_t$ s.t. $\Pr_{\mathcal{D}}\left[\hat{h}_t(\boldsymbol{x}, \boldsymbol{z}_{<t}) \neq z_t\right] \leq \epsilon/T$. Therefore, using the union bound, with probability at least $1 - \delta$, we get:

$$\Pr\left[\exists t \leq T \text{ s.t. } \hat{h}_t(\boldsymbol{x}, \boldsymbol{z}_{<t}) \neq z_t\right] \leq \sum_{t \leq T} \Pr\left[\hat{h}_t(\boldsymbol{x}, \boldsymbol{z}_{<t}) \neq z_t\right] \leq \epsilon$$

$\square$

*Proof of Theorem 6.* By the definition of AR Learnability, we can find a hypothesis $\hat{h}$ s.t., with probability at least $1 - \epsilon$ over $\boldsymbol{x} \sim \mathcal{D}$, we get

$$\hat{h}(\boldsymbol{x}, h^{(1)}(\boldsymbol{x}), \ldots, h^{(t)}(\boldsymbol{x})) = h(\boldsymbol{x}, h^{(1)}(\boldsymbol{x}), \ldots, h^{(t)}(\boldsymbol{x}))$$

for all $t$. So, for such $\boldsymbol{x}$ we get

$$\hat{h}^{(1)}(\boldsymbol{x}) = \hat{h}(\boldsymbol{x}, \emptyset) = h(\boldsymbol{x}, \emptyset) = h^{(1)}(\boldsymbol{x})$$

and by induction:

$$\begin{aligned}
\hat{h}^{(t)}(\boldsymbol{x}) &= \hat{h}(\boldsymbol{x}, \hat{h}^{(1)}(\boldsymbol{x}), \ldots, \hat{h}^{(t-1)}(\boldsymbol{x})) \\
&= \hat{h}(\boldsymbol{x}, h^{(1)}(\boldsymbol{x}), \ldots, h^{(t-1)}(\boldsymbol{x})) \\
&= h(\boldsymbol{x}, h^{(1)}(\boldsymbol{x}), \ldots, h^{(t-1)}(\boldsymbol{x})) = h^{(t)}(\boldsymbol{x})
\end{aligned}$$

$\square$

*Proof of Theorem 8.* Let $f$ be some target circuit, and we define the depth of some gate in the circuit to be the maximal number of nodes in a path connecting the gate to some input variable. We sort the gates in the circuit by their depth, and let $f^{(1)}, \ldots, f^{(T)}$ be the functions computed by the gates in the circuit (where $f^{(T)} = f$ is the output function). Observe that every gate $f^{(t)}$ can be computed by the argmax of a linear function of the inputs and previous gates, and therefore we can define some linear hypothesis $h$ s.t. $h(\boldsymbol{x}, f^{(1)}(\boldsymbol{x}), \ldots, f^{(t-1)}) = f^{(t)}(\boldsymbol{x})$. By induction, we get that for every $t$ we have $h^{(t)} = f^{(t)}$ and therefore the required follows. $\square$

*Proof of Theorem 12.* To show that the length complexity is $O(n/k)$, observe that it is enough to construct a Boolean circuit of size $O(n/k)$, where every gate computes a parity over at most $k$ input bits (similarly to the proof of Theorem 8). This circuit has the structure of a tree, where each node has in-degree at most $k$. It is easy to see that such a tree, with depth $\log_k(n)$ and $O(n/k)$ internal nodes can compute the parity over any subset of bits from the input.

We now show that the length complexity is lower bounded by $\Omega(n/k)$. Assume, for the sake of contradiction, that $\mathcal{P}_n$ can be computed with length complexity $T \leq n/2k$, and particularly this implies that the parity over all input bits (namely, $\chi_{[n]}$) can be computed with $T \leq n/2k$. Observe that, by the choice of the function class, at every step $t$ we have $h^{(t)}(\boldsymbol{x}) = \chi_{A_t}(\boldsymbol{x})$ for some subset $A_t \subseteq [n]$. Additionally, at every step $t$, the size of $A_t$ can increase by at most $k$. Therefore, after $T \leq n/2k$ steps, $h^{(T)}(\boldsymbol{x}) = \chi_{A_T}$ for some $A_T \subsetneq [n]$, and therefore $h^{(T)}(\boldsymbol{x})$ does not compute (or even approximate) $\chi_{[n]}$. $\square$

# B  ADDITIONAL FIGURES

| | |
|---|---|
| **Prompt:** | 1394×8618= |
| **Output (MLP):** | (4×1+9×10+3×100+1×1000)×
(8×1+1×10+6×100+8×1000)=
4×8×1×1+4×1×1×10+4×6×1×100+
4×8×1×1000+9×8×10×1+9×1×10×10+
9×6×10×100+9×8×10×1000+3×8×100×1+
3×1×100×10+3×6×100×100+
3×8×100×1000+1×8×1000×1+
1×1×1000×10+1×6×1000×100+
1×8×1000×1000=
32+040+2400+32000+720+0900+54000+
720000+2400+03000+180000+2400000+
08000+010000+0600000+08000000=
**12013492** |
| **Output (GPT-3.5):** | The multiplication of 1394 by 8618 is
equal to **12** , **013** , **6** **92** . |
| **Output (GPT-4):** | The multiplication of 1394 and 8618
equals **12** , **01** **4** , **05** **2** . |
| **Correct Answer:** | **12013492** |

Figure 3: Comparison between the output of our MLP, GPT-3.5 and GPT-4 on the 4-digit multiplication task.

