# OpenReview forum: "Auto-Regressive Next-Token Predictors are Universal Learners"
_ICLR.cc/2024/Conference — Submitted to ICLR 2024_

### Official Review · Reviewer_ry1v · 2023-10-24

**Soundness:** 3 good
**Presentation:** 3 good
**Contribution:** 3 good
**Rating:** 8
**Confidence:** 3

**Summary:**

This paper introduces a theoretical framework for analyzing autoregressive learners. At its core is a formal definition of autoregressive learnability (AR Learnability), analogous to PAC Learnability.

AR Learnability is defined with respect to distributions over strings over a finite set of tokens $\mathbb{D}$. More specifically, these are distributions $\mathcal{D}$ over contexts $\mathcal{X} = \mathbb{D}^n$ and continuations $\mathcal{Z} = \mathbb{D}^T$ for some integers $n$ and $T$.
Briefly stated, a hypothesis class $\mathcal{H}$ is AR Learnable if there is an algorithm that, for any realizable distribution $\mathcal{D}$, takes as input samples from $\mathcal{D}$ and outputs (w.h.p.) a hypothesis $h \in \mathcal{H}$ such that $h(x,z_{\<t}) = z_t$ for most $t \leq T$. That is, a hypothesis that can (mostly) correctly generate the continuation of the context $x$ as in the distribution $\mathcal{D}$.

This definition is rigorously analyzed. The first batch of theoretical findings (theorems) are concerned with the universality of AR learners:
1. The product of $T$ PAC-learnable hypothesis classes AR-learable [Theorem 3].
2. A generalization of AR-learability to the setting of computing/approximating functions; AR-learnable classes are approximable [Theorem 5]
3. Linear AR models ($\mathcal{H}^{\mathrm{lin}}$) can approximate any $f \in \mathrm{TIME}(T(n))$ w.r.t any distribution $\mathcal{D}$ given access to a dataset of $\mathrm{poly}(\mathrm{T(n)})$-length strings [Corollary 8].

The last item above is particularly significant because, in the supervised (i.e., "standard") setting, the analogous class to $\mathcal{H}^{\mathrm{lin}}$ are known to be so weak that they cannot learn the XOR function. To my understanding, Corollary 8 is proven by representing the function $f$ as a linear threshold circuit $C_f$; the dataset is populated by sampling an $x \gets \mathcal{D}$ and adding a row that represents the outputs of intermediate gates of $C_f(x)$.

Considering the above proof, it is perhaps not surprising that AR learners are universal learners; the "heavy-lifting" is done by the dataset, which verbosely represents the computed function. Thus, the authors define and analyze a new complexity measure for AR learners, called the _length complexity_: For a given hypothesis class $\mathcal{H}$ and a target function class $\mathcal{F}$, the length complexity is the [minimal? see Weaknesses] number of autoregressive steps $T$ such that each $f \in \mathcal{F}$ can be computed  by some $h \in \mathcal{H}$ within $T$ autoregressive steps. Intuitively, length complexity should capture the length of a chain-of-thought used when training the model.

Finally, the authors show that parities over $n$ bits can be computed by $\mathcal{H}^{\mathrm{lin}}$ with length complexity $O(\log n)$. Additionally, they prove a theorem depicting a trade-off between the richness of the hypothesis class and the length complexity, by showing that parities over $n$ bits can be learned by $k$-order parities over $\geq n$ bits, with length complexity $\Theta(n / k)$.

The authors supplement the theoretical framework and findings with experimental evaluation. First, they show that an adaptation of $\mathcal{H}^{\mathrm{lin}}$ with 162M parameters exhibits non-trivial learning on TinyStories; it is often able to generate coherent text. Second, they show that a 4-layer MLP with 775M parameters can learn to multiply 4 digit numbers as well as Goat 7B, provided sufficiently long chain-of-thought (i.e., length complexity) and a custom tokenization scheme.

**Strengths:**

- This paper provides a theoretical foundation for an increasingly important topic, namely, understanding the emergent abilities of autoregressive learners.
- Several theoretical works have already tackled the questions of autoregressive learning. However, to my knowledge, this is the first paper to propose a generic defintion analogous to PAC-learning. This is significant because it may inspire learning theorists to search for analogous results to what is known in the rich literature of PAC. For example, as the authors suggest, it would be interesting to find an intrinsic dimension of hypothesis classes that governs AR-learnability (cf. VC-dimension).
- The theoretical framework presented in this paper is an appropriate operationalization of autoregressive learning: the definitions stay close to autoregressive learning as it is implemented in practice (at least conceptually).
- I find the framework itself appealing. It is not easy to come up with a clean definition based on an existing phenomenon, as they real world is often "messy". The definitions strike a good balance between intuitiveness/simplicitly, to being non-trivial ("mysterious") enough to invite further analysis.
- Overall, the paper is clearly written. Reading it was an enjoyable experience!
- I particularly appreciate how well-organized this paper is. It first presents a fairly intuitive definition, and then provides a sequence of theorems and refinements of this definition to interesting settings. That is to say, the paper "tells a convincing story" about autoregressive learning.
- This "story" that the theory suggests is then supported by experimental results fairly adequetly. While the experimental setup is somewhat simplistic as compared to more empirical works, I do not think that a grander setup is needed for this type of paper.
- While I did not verify the proofs of all theorems, I have read some of them (especially Theorem 7) and they seem correct. I would also like to emphasize that the simplicity of the proofs should not be viewed as a weakness of this paper. Coming up with the right definition (from which intuitive theorems are easy to prove) is a challenging part of many theoretical works; in this case, the authors did well in carefully designing their framework.

**Weaknesses:**

Listed in decreasing order of significance.

## TinyStories experiment is anecdotal, compares to wrong model?
I am not sure what are the actual results being reported with the TinyStories experiment: What I found is a footnote on a 1.2 difference in perplexity between the linear predictor and GPT-2 Small---but I'm not sure what to make of this quantity. And there is a statement that the linear predictor "often does produce coherent text". But how often? And how do you measure coherence? While I lack the expertise to suggest a concrete experiment, I expect a higher level of rigour for a paper at ICLR. Perhaps the authors could refer to the TinyStories paper and reproduce some of the experiments there with their model.

Relatedly, it seems that in the TinyStories paper there is a 28M-parameter model that achieves performance comparable to GPT2-XL. Therefore, the comparison of the linear predictor to GPT2-Small seems inappropriate---much better performance can be attained with 5x less parameters. Perhaps the authors should compare the linear predictor to [that model](https://huggingface.co/papers/2305.07759) instead. Discovering that a linear predictor requires significantly more parameters than a (good) transformer model to achieve comparable performance would undermine the main message of the paper (that the power of LLMs can be attributed to AR rather than architecture). Please correct me if you disagree that comparing to the TinyStories transformer is a more fair comparison towards this end.

## Linear Decoders should be defined more slowly
- Given the significance of linear ARs (linear decoders) throughout the paper, I suggest defining them more slowly and in their own Definition environment---rather than presenting them as an "Example". A Definition environment is more easy to refer back to, and encourages more formal writing.
- A figure depicting the construction would be welcome: I had to work it out with pen and paper on the margin.
- "Under some margin conditions and using a convex surrogate loss function, this class is in fact learnable using SGD." This sentence must be supported by either a citation or a proof. As it is currently phrased it is too vague and not well-enough argued for to be used as a true statement (the current level of rigour is more appropriate for a tangential side-note).
- Observe that this class is learnable in polynomial time: Say PAC-learnable, becuase this paper uses multiple notions of learnability.

## Length complexity is not well-defined
As length complexity is currently defined in Definition 9, a class $\mathcal{H}$ has infinitely many length complexities for computing a given $\mathcal{F}$. This is because if $T$ is a length complexity of $\mathcal{H}$ for computing $\mathcal{F}$, then so is any $T' \geq T$. Instead, you should define the length complexity of $\mathcal{H}$ computing $\mathcal{F}$ to be _the minimal $T$ for which the conditions stated in the definition hold.

On that note, it would be illuminating (and add to the cohesion of the paper) if the authors spell-out the length complexity of the construction from Theorem 7.

## Missing citations
- Towards Revealing the Mystery behind Chain of Thought: A Theoretical Perspective by Feng et al. (2023): This work gives a complexity-theoretic explanation to the success of chain-of-thought, especially in mathematical reasoning. The techniques seem entirely different, and it is focused on transformer architectures (rather than focusing on the autoregressive aspect of LLMs, as in the paper currently under review). Still, it clearly fits in three of the four topics covered by the related work (all save for "Beyond Transfomers").
- Fast Learning Requires Good Memory: A Time-Space Lower Bound for Parity Learning by Raz (2016): This prominent result (FOCS best paper, JACM) should probably be mentioned in the first paragraph of Section 2.3.1 that discusses hardness results for parities.

## Other minor writing comments
- Page 4 above Definition 2: "...for all **sub-sequences** $z_{<t}$" should be **prefixes** (sub-sequences is correct but less accurate).
- Definition 2: Should say "for every $\epsilon, \delta \in (0,1)".
- Definition 2: Should say "If there exists $m \colon (0,1)^2 \to \mathbb{N}$" such that... Otherwise, the order of quantifiers / role of $m$ is unclear.
- Definition 2: Should say "returns w.p. $\geq 1-\delta$ a function $h \in \mathcal{H}$. Also, the first use of w.p. should be explicitly defined ("with probability (w.p.)") so that the paper is accessible to a broad audience.
- Theorem 3: It would be nice to write "then \mathcal{H} = \mathcal{H}_1 \times \dots \times \mathcal{H}_T". Ideally, theorem statmenets should be as self-contained as possible as they are often used for quick reference.
- Definition 4: I would say that "$h$ computes $f$ w.r.t \mathcal{D}". That is, explicitly include $\mathcal{D}$ in the definition, since it is crucial. Likewise for the definition of _approximates_.
- Theorem 5: If the previous comment is accepted, this statement should be updated.
- Theorem 6: If my understanding of the proof is correct (see Summary of this review, above), it is worth adding a sentence explaining what the dataset is to the body of the paper---rather than just saying the main fact the proof relies on, explain how it is used.
- Theorem 7 and Corollary 8: use the notation $\mathcal{H}^{\mathrm{lin}}$ here when referring to linear AR functions/models. Otherwise, the notation surprisingly re-appears on the next page.
- Section 2.3.1: This is a matter of taste, but I would find it more informative to use $\mathcal{P}_n$ to denote the class of parities on $n$ bits. This is because $\mathcal{F}$ was previously used to refer to a generic class of functions.
- Section 2.3.1, second paragraph after Theorem 10: when defining $\mathcal{F}_{n,k}$, it should be $A \in \binom{[n]}{\leq k}$. Note the square brackets around $n$.
- Section 2.3.1, above Theorem 11: It would be much better to avoid using $\approx$ notation in favor of notation with more well-defined meaning, such as $O(\cdot)$.
- Section 2.3.1, above Theorem 11: Should be "and a sample complexity of $\approx k \log n$" (not "the sample complexity").

**Questions:**

Many of the suggestions/qualms I listen in Weaknesses above can be rephrased as a question. I am open to discussing any of these, and would consider updating my score based on these being addressed in a revision. My only remaining question is whether the authors intend to release the code used for their experiments so that the reader may reproduce them.

---

> ### Author Response · Authors · 2023-11-15
>
> Thank you very much for your overall positive and encouraging feedback, and thank you for acknowledging the novelty of our theoretical framework. We find your detailed comments and remarks to be very valuable for improving the paper.
>
> We respond to the main weaknesses raised in the review:
> - TinyStories experiment: following your remarks, we will conduct and report additional thorough and rigorous experiments on the TinyStories dataset in the final version of the paper. We will use some of the experimental settings in the original TinyStories paper to provide more quantitative results on the TinyStories dataset. We will also compare our linear model to the smaller transformer models studied in the TinyStories paper. However, note that we do not claim that linear models are better than transformers for language modeling, and certainly they can be much less efficient in terms of parameters. The primary reason for studying linear auto-regressive models is because we can achieve a more complete theoretical understanding of their optimization and expressive power, something that we are not able to do for Transformers. Hence, the TinyStories experiment is mainly used for demonstrating that linear models can generate reasonable text and give non-trivial performance, a result that we found surprising.
> - Definition of linear decoders: we have updated the paper to reflect your comments, giving a more structured introduction of linear decoders.
> - Length complexity definition: we updated the paper to clarify the definition of length complexity.
> - Missing citations: we added the citations you suggested to the updated manuscript.
> - Other minor writing comments: we addressed all the comments in the updated paper. Thank you again for these detailed and valuable suggestions.
>
> We believe the updated manuscript addresses most of the concerns raised in the review. We will add further experiments to the final paper. We also plan to publish the code for reproducing all the experiments along with the final paper. We are happy to address any additional concerns you have regarding the paper, and would appreciate your feedback on the updated manuscript. Please consider updating your score if your concerns were addressed.

---

> > ### Comment · Reviewer_ry1v · 2023-11-21
> >
> > Thank you. The revision addresses all of my concerns except for my most major one, regarding the TinyStories experiment. Based on the assumption you will indeed overhaul this section as mentioned in your comment, I have increased my overall score.

---

### Official Review · Reviewer_QtWu · 2023-10-28

**Soundness:** 2 fair
**Presentation:** 3 good
**Contribution:** 2 fair
**Rating:** 5
**Confidence:** 3

**Summary:**

The paper introduces a theoretical framework for studying auto-regressive next-token prediction models. The key idea is that auto-regressive learning, where the model is trained to predict the next token given previous tokens, allows the model to utilize intermediate "chain-of-thought" computations. This gives the model more information compared to standard supervised learning where only input-output pairs are provided.

The authors show theoretically that even simple auto-regressive models like linear next-token predictors can approximate any efficiently Turing-computable function, if provided with appropriate intermediate chain-of-thought supervision. They also introduce a new notion of "length complexity" which measures the number of intermediate tokens needed to learn a target function. Length complexity allows trading off sample/computational complexity and length complexity.

Experiments on text generation and arithmetic tasks demonstrate that simple models like linear networks and small MLPs can perform well when trained with next-token prediction and chain-of-thought supervision.

**Strengths:**

(1) Provides an elegant theoretical framework for studying auto-regressive next-token prediction models, an important class of models in NLP.

(2) Establishes strong learnability and approximation guarantees for simple models like linear predictors when trained auto-regressively.

(3) Introduces the novel concept of "length complexity" to capture chain-of-thought requirements. Relates length complexity to sample and computational complexity.

**Weaknesses:**

(1) The theoretical results rely on very strong assumptions about availability of chain-of-thought training data, which may be unrealistic.

(2) More analysis would be useful on how length complexity scales with problem complexity for different hypothesis classes.

(3) Additional validation on more complex architectures like Transformers would strengthen the conclusions about training scheme vs architecture.

(4) The proposed linear models are not exactly equivalent to the classical linear models analyzed. Some parameters are shared across time steps.

(5) Non autoregressive models are not discussed, tested and compared. It will be to know if given large amount of training data, non-autoregressive models can perform on par with autoregressive models.

(6) On MULTIPLICATION experiments, models are trained and tested on 4-digit numbers. The model is likely to memorize the patterns instead of generalizations, especially this paper use special tokenization for digits and signs.

**Questions:**

This paper misses some critical references:

(1) Training of smaller student models on CoT data has been investigated in several earlier papers [1, 2].
(2) Language Models for Arithmetic Tasks have been extensively discussed and studied in [3].



[1] Li et al. Explanations from Large Language Models Make Small Reasoners Better. 2022.
[2] Magister et al.Teaching Small Language Models to Reason. ACL, 2023.
[3] Qian et al. Limitations of Language Models in Arithmetic and Symbolic Induction. ACL, 2023.

---

> ### Author Response · Authors · 2023-11-15
>
> Thank you very much for your comments and valuable feedback.
> Regarding the weaknesses raised in the review:
>
> 1. Indeed our theoretical results rely on availability of chain-of-thought training data. We note that many tasks do have CoT data for training (or fine-tuning), and that training on data with CoT demonstrations was found to greatly improve performance. For example, the GSM8K dataset contains CoT answers, and recent works suggest that augmenting the training data with different forms of CoT dramatically improves performance (e.g. https://arxiv.org/pdf/2309.12284.pdf). We do agree however that our theoretical results rely on availability of “unrealistic” CoT data, and we acknowledge this in the paper as well. As is often the case in many theoretical works, we study an idealized setting that allows us to give concrete analysis of specific simple models, studying their capabilities and limitations. We believe that our main theoretical result in this setting, showing that simple models such as linear autoregressive predictors can learn to compute any (computable) function with CoT, is surprising and novel.
>
> 2. Our work provides a novel theoretical framework for studying the interaction between problem complexity and length complexity, and we derive some preliminary analysis of length complexity for specific problems. Since our main contribution in this work is in introducing such a theoretical framework, we leave a further investigation to future work.
>
> 3. Following your suggestion, we will add additional experiments comparing non-autoregressive to autoregressive Transformers in the final version of the paper.
>
> 4. Indeed, the linear model trained in our experiments is in fact a linear network, which is somewhat different from the models we analyze. However, we note that the overall model in the experiment still computes a linear function of the inputs. In other words, the models used in the experiments are still linear auto-regressive models (as defined in the theory section), but they are trained with a linear-network parameterization which enforces some weight sharing.
>
> 5. Our theoretical results imply that, at least for linear models, non-autoregressive models are significantly weaker than autoregressive models. This is true regardless of the amount of training data used, since non-autoregressive linear models cannot express rich functions that autoregressive linear models can. We note that very recently, a similar theoretical result was shown for the Transformer architecture (https://arxiv.org/pdf/2310.07923.pdf).
>
> 6. Generalization on the multiplication experiment: note that we split the data to train and evaluation pairs, and report the performance when multiplying pairs of numbers that were never seen during training. Therefore, the model generalizes to unseen pairs, applying the same algorithm learned during training.
>
> We added additional references following the suggestions in the review. We note that we already discuss Qian et al., 2023 in the original manuscript.
>
> We are happy to answer any further questions. Please consider raising your score if you believe your questions were answered.

---

### Official Review · Reviewer_RXBR · 2023-11-01

**Soundness:** 2 fair
**Presentation:** 3 good
**Contribution:** 2 fair
**Rating:** 5
**Confidence:** 4

**Summary:**

This paper attempts to provide a theoretical explanation of how auto-regressive prediction in LLMs can enable them to perform nontrivial tasks.

The theoretical idea is that a dataset including intermediate steps can enable simple (in fact, linear) next-token predictors to learn to perform arbitrary computable functions.

The paper develops a theory formalizing these ideas.
It first presents a notion of autoregressive learnability modeled on PAC learnability.
It then derives results on the approximability of problems with autoregressive predictors -- that is, in the presence of intermediate steps. Indeed, for any Turing-computable function, a linear autoregressive model can compute this function, with a number of intermediate steps polynomial in the Turing machine's runtime.
It finally investigates the number of required intermediate steps, at the example of k-th order parities in length-n strings. This "length complexity" can trade off with the computational complexity, as a shorter sequence may require more complex individual steps.

It then presents two experiments showing that simple autoregressive predictors can perform nontrivial tasks.

The first experiment qualitatively shows that a linear autoregressive model trained on TinyStories can produce mostly-reasonable-looking text.
The second experiment shows that, with appropriate intermediate steps and an adapted input formatting, an MLP-based autoregressive model can multiply 4-digit numbers.

**Strengths:**

- Provides an interesting angle on the success of chain-of-thought in enabling LMs to perform more complex tasks

- provides both theoretical analysis and empirical evidence

**Weaknesses:**

- The TinyStories experiment (Section 3.1) lacks quantitative evaluation. It is unclear if the examples shown are representative.

- Multiplication experiment: unlike the TinyStories experiment, the model is not linear -- is this important? Why not use the same model for both experiments?

- There appears to be a potential mismatch with realistic chain-of-thought prompting in that the learnability theory developed here assumes that the tasks are available together with their full sequences of intermediate steps in the training set, whereas in reality prompts containing chain-of-thought demonstrations may be quite unnatural, and are quite likely not to have appeared in that form in the training set. In this sense, the learnability theory, if understood as applying to unsupervised LLM training, does not explain why LMs would be able to deal with unnatural prompt formats.

- In the multiplication experiment, the calculations are decomposed into "more intermediate steps than in Liu&Low". Furthermore, they involve padding to get all strings to have the same length, apparently unlike Liu&Low. Both of these steps may impact the comparison with the results from Liu&Low. Furthermore, unlike the MLP, GPT-3.5 is (judging by Figure 2) not prompted to output intermediate steps. Thus, it is unclear what one can learn from the comparison between the models in Figure 2 (right), and the statement "outperforms GPT-4" in the introduction may not be fully supported.

**Questions:**

- Section 3.1: what is the objective function -- is it cross-entropy? Is softmax applied on the linear output?

- Section 2.1, first paragraph: Why is \mathcal{D} a distribution -- given that its support is finite and that the only probabilistic statement made then holds with probability 1 (next line), could \mathcal{D} just as well be a subset of X \times Z_T?

- Definition 2: when encountering this, I wondered: is T a global free variable (i.e., AR-Learnability is defined w.r.t. T), or is T dependent on \mathcal{D}? This is resolved in Footnote 2. Maybe disambiguate this at the beginning of Section 2.1?

- minor: "Multi-Linear Perceptron (MLP)" (page 2, third paragraph) --> the standard reading of "MLP" appears to be Multi-Layer Perceptron, and indeed the model appears to be of this type

---

> ### Author Response · Authors · 2023-11-15
>
> Thank you very much for your comments and valuable feedback.
> Regarding the weaknesses raised in the review:
> - TinyStories experiments: We will conduct and report additional thorough and rigorous experiments on the TinyStories dataset in the final version of the paper, and add additional prompts and completions. We do note that the examples shown in the paper are representative and are not “cherry-picked” examples.
> - Multiplication experiment: Indeed, we use an MLP instead of a linear network for the multiplication experiment, as we found them to give more consistent performance. We do note that these models are still extremely simple compared to the standard transformer architecture.
> - Availability of chain-of-thought during training: We agree that in some cases, chain-of-thought reasoning for specific tasks does not appear in the training set. However, note that many tasks do have CoT data for training (or fine-tuning), and that training on data with CoT demonstrations was found to greatly improve performance. For example, the GSM8K dataset contains CoT answers, and recent works suggest that augmenting the training data with different forms of CoT dramatically improves performance (e.g. https://arxiv.org/pdf/2309.12284.pdf). Our work focuses on such settings, where CoT is available as part of the training data.
> - More intermediate steps in the multiplication experiment: indeed, our model is trained with longer chain-of-thought data compared to the paper of Lie&Low, and is also compared to GPT-3.5/4 which do not use CoT in their answer. On the other hand, in our work we use an extremely simple architecture (shallow MLP) compared to the gigantic transformers in competing methods. Our goal in our experiment is not to convince readers that MLPs or linear networks are better than transformers - they clearly are not. Our only goal is to emphasize that longer CoT data can make such simple models competitive with much larger and more complex networks. In other words, instead of adding complexity to the model, we are adding complexity to the CoT training data.
>
> Regrading the questions raised in the review:
> - Section 3.1: yes, we train with standard cross-entropy loss, with softmax over the outputs. We clarify this in an updated version of the manuscript.
> - Section 2.1: note that we use the fact that $\mathcal{D}$ is a distribution to define AR learnability using a sample from $\mathcal{D}$  (Definition 2).
> - Definition 2: thank you for this comment. In the updated manuscript we define the variable T earlier in the paragraph to make the definition clearer.
> - We fixed the typo in the definition of MLP.
>
> We are happy to answer any further questions. Please consider raising your score if you believe your concerns were answered.

---

> > ### Comment · Reviewer_RXBR · 2023-11-18
> >
> > Thanks for the response. The Questions are all answered to my satisfaction.
> > Taking your word on the Tinystories experiment, I see the first one of the Weaknesses as addressed. While I understand some of the motivation for what you did regarding the other Weaknesses as explained in the response, I still maintain that the third and fourth points under "Weaknesses" are weaknesses of the paper.

---

### Meta-Review · Area_Chair_s78B · 2023-12-06

**Metareview:**

This paper introduces a theoretical framework for studying auto-regressive next-token prediction models. By predicting the next token given previous tokens, the authors show that the model can utilize intermediate "chain-of-thought" computations to complete challenging tasks.

The work is interesting in general. However, the reviewer raised multiple concerns, including strong assumptions (training data construction and linear model) and experimental design (multiplication experiment and more baselines). Both of the concerns are critical and are not well-addressed during the rebuttal phase. Given those problems, I recommend rejection and hope the authors address the problems and resubmit the work to a future venue.

**Justification For Why Not Higher Score:**

See the metareview

**Justification For Why Not Lower Score:**

N/A

---

### Decision · Program_Chairs · 2024-01-16

Reject